# A loop structure allows TAPBPR to exert its dual function as MHC I chaperone and peptide editor

**Lina Sagert, Felix Hennig, Christoph Thomas\*, Robert Tampé\***

Institute of Biochemistry, Biocenter, Goethe University Frankfurt, Frankfurt, Germany

**Abstract** Adaptive immunity vitally depends on major histocompatibility complex class I (MHC I) molecules loaded with peptides. Selective loading of peptides onto MHC I, referred to as peptide editing, is catalyzed by tapasin and the tapasin-related TAPBPR. An important catalytic role has been ascribed to a structural feature in TAPBPR called the scoop loop, but the exact function of the scoop loop remains elusive. Here, using a reconstituted system of defined peptide-exchange components including human TAPBPR variants, we uncover a substantial contribution of the scoop loop to the stability of the MHC I-chaperone complex and to peptide editing. We reveal that the scoop loop of TAPBPR functions as an internal peptide surrogate in peptide-depleted environments stabilizing empty MHC I and impeding peptide rebinding. The scoop loop thereby acts as an additional selectivity filter in shaping the repertoire of presented peptide epitopes and the formation of a hierarchical immune response.

**\*For correspondence:**
c.thomas@em.uni-frankfurt.de (CT);
tampe@em.uni-frankfurt.de (RT)

**Competing interests:** The authors declare that no competing interests exist.

## Introduction

Nucleated cells of higher vertebrates provide information about their health status by presenting a selection of endogenous peptides on MHC I molecules at the cell surface. By sampling these peptide-MHC I (pMHC I) complexes, CD8+ T lymphocytes are able to detect and eliminate infected or cancerous cells (*Blum et al., 2013*; *Rock et al., 2016*). In a process called peptide editing or proofreading, peptides derived from the cellular proteome are selected for their ability to form stable pMHC I complexes. This peptide editing is known to be catalyzed by the two homologous MHC I-specific chaperones tapasin (Tsn) and TAP-binding protein-related (TAPBPR) (*Fleischmann et al., 2015*; *Hermann et al., 2015*; *Morozov et al., 2016*; *Neerincx and Boyle, 2017*; *Tan et al., 2002*; *Thomas and Tampé, 2019*; *Wearsch and Cresswell, 2007*; *Wearsch et al., 2011*). The selection of high-affinity MHC I-associated peptide epitopes is of pivotal importance not only for immunosurveillance by effector T lymphocytes, but also for priming of naïve T cells and T cell differentiation. As an integral constituent of the peptide-loading complex (PLC) in the endoplasmic reticulum (ER) membrane, the ER-restricted Tsn functions in a 'nanocompartment' characterized by a high concentration of diverse, optimal peptides. The peptides are shuttled into the ER by the heterodimeric ABC (ATP-binding cassette) transporter associated with antigen processing TAP1/2, the central component of the PLC (*Abele and Tampé, 2018*). In the ER, most peptides are further trimmed by the aminopeptidases ERAP1 and ERAP2 to an optimal length for binding in the MHC I groove (*Evnouchidou and van Endert, 2019*; *Hammer et al., 2007*). In contrast to Tsn, TAPBPR operates independently of the PLC and is also found in the peptide-depleted *cis*-Golgi network (*Boyle et al., 2013*). Fundamental insights into the architecture and dynamic nature of the Tsn-containing PLC have come from a recent cryo-EM study of the fully-assembled human PLC (*Blees et al., 2017*), while the basic principles underlying catalyzed peptide editing have been elucidated by crystal structures of the TAPBPR-MHC I complex (*Jiang et al., 2017*; *Thomas and Tampé, 2017a*): TAPBPR stabilizes the peptide-binding

**eLife digest** Cells in the body keep the immune system informed about their health by showing it fragments of the proteins they have been making. They display these fragments, called peptides, on MHC molecules for passing immune cells to inspect. That way, if a cell becomes infected and starts to make virus proteins, or if it becomes damaged and starts to make abnormal proteins, the immune system can 'see' what is happening inside and trigger a response.

MHC molecules each have a groove that can hold one peptide for inspection. For the surveillance system to work, the cell needs to load a peptide into each groove before the MHC molecules reach the cell surface. Once the MHC molecules are on the cell surface, the peptides need to stay put; if they fall out, the immune system will not be able to detect them. The problem for the cell is that not all peptides fit tightly into the groove, so the cell needs to check each one before it goes out. It does this using a protein called TAPBPR.

TAPBPR has a finger-like structural feature called the "scoop loop", which fits into the end of the MHC groove while the molecule waits for a peptide. It was not clear, however, what this loop actually does. To investigate, Sagert et al. mutated the scoop loop of TAPBPR to see what happened to MHC loading in test tubes.

The experiments revealed that the scoop loop plays two important roles. The first is to keep the MHC molecule stable when it is empty, and the second is to hinder unsuitable peptides from binding. The scoop loop sticks into one side of the groove like a tiny hairpin, so that pushed-out, poorly fitting peptides cannot reattach. At the same time, it holds the MHC molecule steady until a better peptide comes along and only releases when the new peptide has slotted tightly into the groove.

Understanding how cells choose which peptides to show to the immune system is important for many diseases. If cells are unable to find a suitable peptide for a particular illness, it can stop the immune system from mounting a strong response. Further research into this quality control process could aid the design of new therapies for infectious diseases, autoimmune disorders and cancer.

groove in a widened conformation primarily through the MHC I α2–1 helix, distorts the floor of the binding groove, and shifts the position of β2-microglobulin (β2m). Furthermore, one of the two TAPBPR-MHC I complex structures revealed a remarkable structural feature in TAPBPR named the scoop loop (*Thomas and Tampé, 2017a*). In TAPBPR, this loop is significantly longer than the corresponding region in Tsn, which was not resolved in the X-ray structure of Tsn (*Dong et al., 2009*). Notably, the scoop loop of TAPBPR is located in the F-pocket region of the empty MHC I binding groove (*Figure 1A,B*). By anchoring the C-terminal part of the peptide, the F pocket region is crucially involved in defining pMHC I stability (*Abualrous et al., 2015*; *Hein et al., 2014*). The scoop loop occupies a position that is incompatible with peptide binding and displaces or coordinates several key MHC I residues responsible for binding the C terminus of the peptide. We therefore proposed that the scoop loop can be regarded as a surrogate for the C terminus of the displaced peptide, stabilizing the inherently labile empty MHC I molecule (*Thomas and Tampé, 2017a*). At the same time, by occupying a region critical to peptide binding, the scoop loop might allow only high-affinity peptides to re-enter the MHC I binding groove after displacement of sub-optimal peptide. The proposed importance of the scoop loop for TAPBPR function has recently been scrutinized in a study by Ilca et al. investigating TAPBPR scoop-loop variants using immunopeptidomics and cell-based assays (*Ilca et al., 2018*). Ilca et al. found that a specific leucine residue in the scoop loop facilitates peptide displacement on MHC I allomorphs favoring hydrophobic peptide side chains in their F pocket. Here, we aimed to clarify the role of the scoop loop during TAPBPR-catalyzed peptide editing using in vitro interaction and peptide-exchange studies with defined, purified components. We demonstrate that the scoop loop is of critical importance for TAPBPR-mediated stabilization of empty MHC I clients in peptide-depleted environments and contributes to peptide quality control during editing by impeding released peptide to rebind in the MHC I groove. Collectively, our data support a crucial role for the TAPBPR scoop loop in establishing a hierarchical immune response.

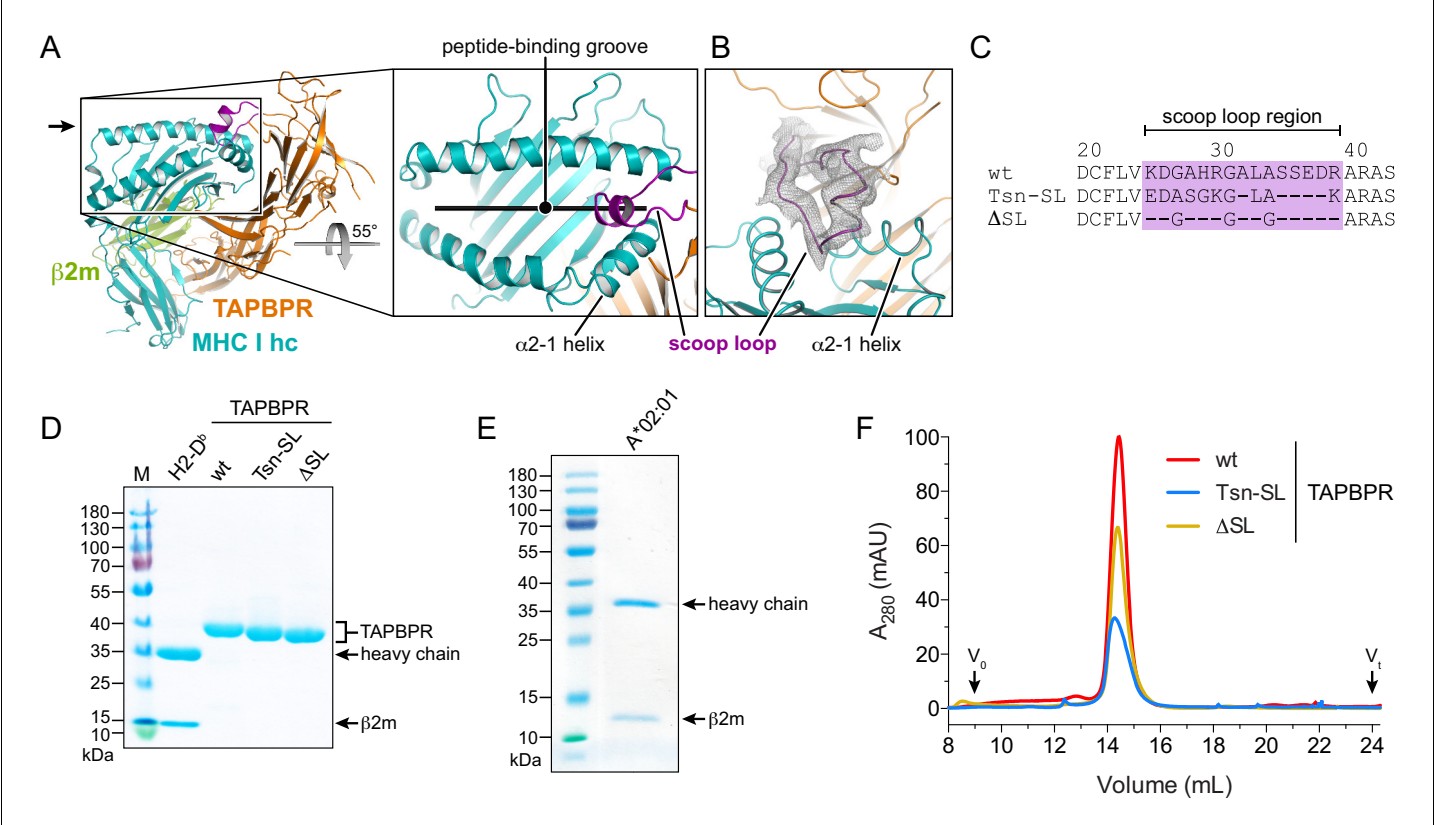

**Figure 1.** Expression and purification of different TAPBPR scoop-loop variants and MHC I chaperone clients. (**A**) X-ray structure of the TAPBPR-MHC I complex in cartoon representation (PDB ID: 5OPI). The zoom-in shows how the TAPBPR scoop loop (purple) is inserted into the F-pocket region of the MHC I peptide-binding groove that is occupied by the C terminus of the peptide before peptide displacement. (**B**) $2F_o$-$F_c$ electron density of the X-ray structure in the region of the scoop loop, contoured at 0.8σ. The width of the helix cartoons has been reduced to facilitate visualization of the electron density. The viewing direction is indicated by the black arrow in panel (**A**). (**C**) Sequence alignment of the scoop-loop region in the TAPBPR constructs used in this study. (**D, E**) Purified proteins used in the current study were analyzed by non-reducing SDS-PAGE. The MHC I allomorphs H2-D[b] (mouse) and HLA-A*02:01 (human) were refolded in the presence of β2m and peptide. (**F**) The TAPBPR proteins, injected at different concentrations to facilitate comparison, eluted as monodisperse samples during size-exclusion chromatography (SEC). Abbreviations: MHC I hc: MHC I heavy chain; wt: wildtype; Tsn: tapasin; SL: scoop loop; M: protein marker; kDa: kilodalton; $A_{280}$: absorption at 280 nm; $V_0$: void volume; $V_t$: total volume.

# Results

## Design of TAPBPR scoop-loop variants

To investigate the function of the scoop loop, we prepared two human TAPBPR variants: TAPBPR[Tsn-SL], in which the TAPBPR scoop loop was replaced with the corresponding shorter loop of Tsn, and TAPBPR[ΔSL], in which the original scoop loop was essentially deleted by replacing it with three glycine residues to preserve proper folding of the MHC I chaperone (*Figure 1C*). The ER-lumenal domains of wildtype (wt) TAPBPR and the variants, each harboring a C-terminal histidine tag, were expressed in insect cells and purified from the cell culture supernatant via immobilized-metal affinity chromatography (IMAC) and size-exclusion chromatography (SEC). As MHC I chaperone clients, we chose mouse H2-D[b] and human HLA-A*02:01, which are known to interact with TAPBPR (*Hermann et al., 2013*; *Ilca et al., 2019*; *Morozov et al., 2016*). HLA-A*02:01, the major MHC I allomorph in the Caucasian population and found in more than 50% of the global population, presents a diverse spectrum of immunodominant autoimmune, viral, and tumor epitopes and is therefore medically highly relevant (*Boucherma et al., 2013*). The MHC I allomorphs were expressed in *E. coli* as inclusion bodies and refolded in the presence of β2m and fluorescently-labeled or

photo-cleavable peptide (*Rodenko et al., 2006*). The highly pure TAPBPR variants and pMHC I complexes eluted as monodisperse samples at expected size during SEC (*Figure 1D–F*).

## Scoop-loop variants have reduced chaperone activity towards peptide-free MHC I

During peptide exchange, MHC I molecules go through a peptide-free high-energy intermediate state after peptide release and before re-entry of a new peptide. A hallmark of peptide editors like TAPBPR is their ability to recognize and chaperone this intermediate until it is located in a peptide-rich environment where a high-affinity peptide ligand can enter the MHC I binding groove (*Thomas and Tampé, 2019*; *Thomas and Tampé, 2017b*). To scrutinize the role of the scoop loop in chaperoning empty MHC I, we tested the ability of our TAPBPR variants to stabilize peptide-free H2-D$^b$. Hence, H2-D$^b$ (10 µM) loaded with a photo-cleavable peptide was incubated with TAPBPR (3 µM) under UV exposure. Subsequent SEC analysis revealed that both TAPBPR$^{Tsn-SL}$ and TAPBPR$^{\Delta SL}$ are, in principle, competent to form complexes with MHC I (*Figure 2A*). However, in comparison to TAPBPR$^{wt}$ (*Figure 2A,B*), the amount of H2-D$^b$ complex detected for TAPBPR$^{Tsn-SL}$ and TAPBPR$^{\Delta SL}$ during SEC was reduced by around 40% and 90%, respectively (*Figure 2C*). After reanalysis of the MHC I chaperone complexes by SEC, the mutant complexes were mostly dissociated, indicating kinetic instability (*Figure 2—figure supplement 1A*). In contrast, isolation and reinjection of the wt complex showed that it remained stable for the duration of the experiment (*Figure 2—figure supplement 1A,B*). Yet, in the presence of a high-affinity peptide, even the TAPBPR$^{wt}$-MHC I complex dissociated, in accordance with the role of TAPBPR as a chaperone that stabilizes the MHC I as long as no optimal peptide is present (*Figure 2—figure supplement 1B*). Taken together, these findings demonstrate that the scoop loop is crucial to an extended lifetime of the chaperone-client complex, enabling the escorting of empty MHC I by TAPBPR in a peptide-deficient environment.

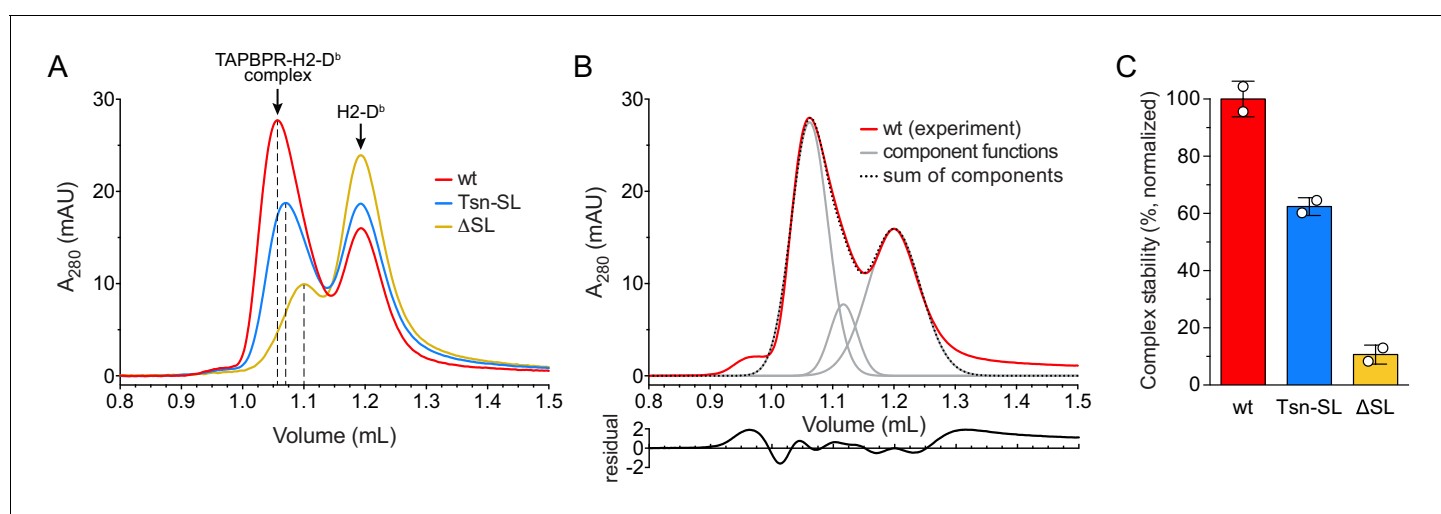

**Figure 2.** Complex formation between MHC I and TAPBPR variants. (**A**) H2-D$^b$ (10 µM) loaded with a photo-cleavable peptide (RGPGRAFJ*TI, J* denotes photocleavable amino acid) was irradiated with UV light in the presence of TAPBPR$^{wt}$ (3 µM, red), TAPBPR$^{Tsn-SL}$ (blue), or TAPBPR$^{\Delta SL}$ (yellow) and subsequently analyzed by SEC. The different elution volumes of the first main peak, marked by dashed lines, already hint at different complex stabilities. (**B**) Deconvolution of size-exclusion chromatogram from TAPBPR$^{wt}$ complex formation (experiment independent of the sample shown in (**A**)). The experimental chromatogram (red) was deconvoluted using three Gaussian functions (gray) that can be ascribed to the TAPBPR-H2-D$^b$ complex (1.06 mL), free TAPBPR (1.12 mL), and free H2-D$^b$ (1.20 mL). The sum of the three Gaussians is shown as dotted curve. The residual plot depicted beneath the main panel shows the difference between the experimental data and the sum. (**C**) Stability of complexes formed by TAPBPR$^{wt}$, TAPBPR$^{Tsn-SL}$, and TAPBPR$^{\Delta SL}$, respectively, as judged by the area of the complex peak obtained by deconvolution. Data represent mean ± SD (n = 2).

The online version of this article includes the following figure supplement(s) for figure 2:

**Figure supplement 1.** Stability of the MHC I complex formed by TAPBPR$^{wt}$ and the TAPBPR scoop-loop variants.

## Scoop-loop variants retain their function in catalyzing peptide dissociation from MHC I

After investigating the chaperone activity of the TAPBPR scoop-loop mutants, we tested their ability to displace MHC I-bound peptide. To this end, we employed an in-vitro peptide exchange assay similar to the one previously described for measuring the activity of Tsn (*Fleischmann et al., 2015*; *Chen and Bouvier, 2007*). Dissociation of medium-affinity fluorescent peptide from refolded and purified p\*MHC I (p\* denotes fluorescently-labeled peptide) was monitored by fluorescence polarization after addition of a 1000-fold molar excess of unlabeled high-affinity competitor peptide in the absence or presence of TAPBPR (*Figure 3A*). The large molar excess of unlabeled competitor peptide ensures that once a fluorescent peptide dissociates, it does not rebind, but is replaced by an unlabeled competitor-peptide molecule. The observed rate constant is thus solely determined by the dissociation rate constant of the fluorescent peptide. The condition of this assay mimics the environment of the PLC, where optimal, high-affinity peptides abound. For the mouse MHC I allomorph H2-D$^b$, TAPBPR$^{wt}$ and the scoop-loop variants accelerated the uncatalyzed peptide release ($2.53 \pm 0.37 \times 10^{-3}$ s$^{-1}$) to a similar extent. The TAPBPR$^{\Delta SL}$ mutant lacking the entire scoop loop

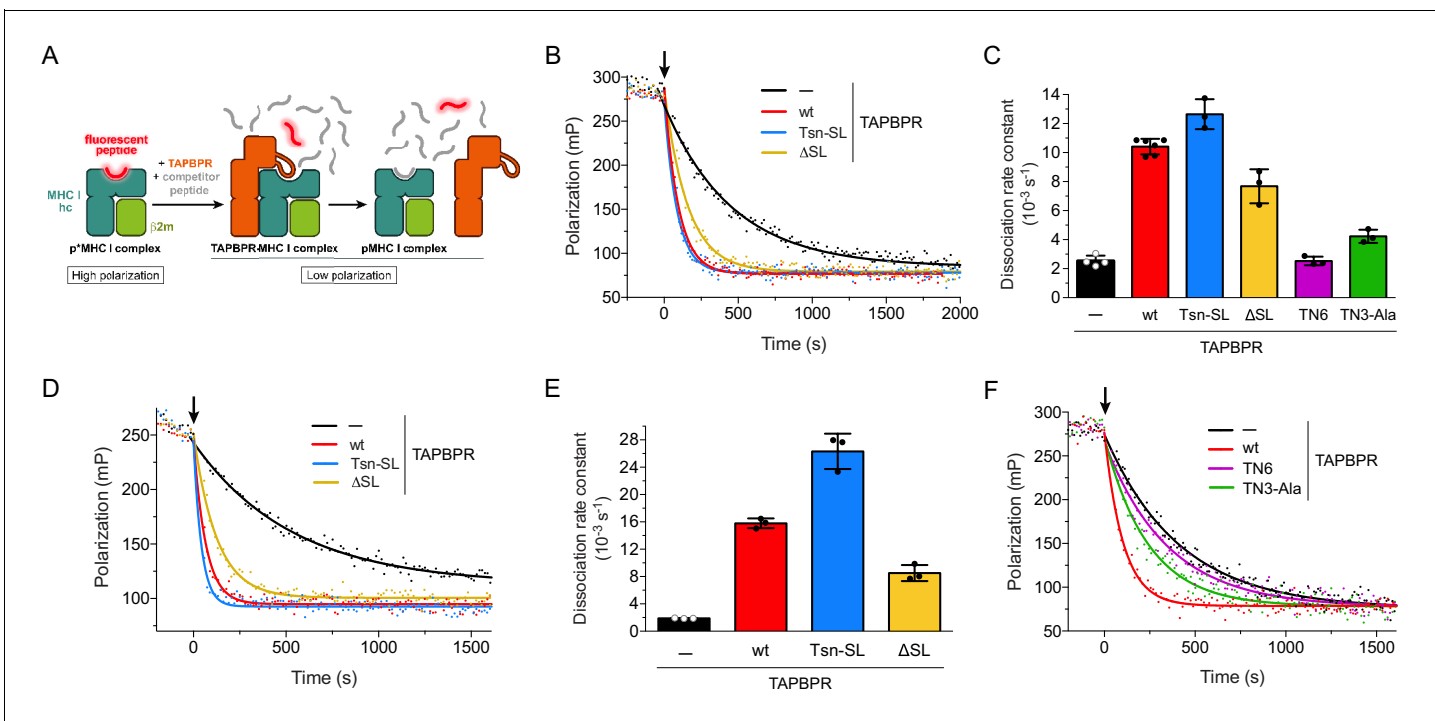

**Figure 3.** Peptide release from H2-D$^b$ and HLA-A\*02:01 in peptide-rich environment. (**A**) Schematic of peptide displacement assay. (**B**) Peptide dissociation kinetics from H2-D$^b$ (300 nM) loaded with fluorescently-labeled peptide (TQSC\*NTQSI) was monitored in real time by fluorescence polarization. The arrow indicates the addition of a 1000-fold molar excess of unlabeled high-affinity competitor peptide (ASNENMETM) without TAPBPR (black trace) or in combination with 1 μM TAPBPR (red, blue, and yellow traces). (**C**) Average dissociation rate constants of uncatalyzed and catalyzed peptide dissociation from H2-D$^b$, using the same conditions as in (**B**). Data represent mean ± SD (n = 2–6). (**D**) Representative fluorescence polarization traces of uncatalyzed and catalyzed peptide (FLPSDC\*FPSF) dissociation from HLA-A\*02:01 (300 nM). The arrow indicates the addition of a 1000-fold molar excess of unlabeled competitor peptide (FLPSDEEPYV, 300 μM) with and without TAPBPR (1 μM). (**E**) Average dissociation rate constants of uncatalyzed and catalyzed peptide dissociation from HLA-A\*02:01, using the same experimental conditions as in (**D**). Data represent mean ± SD (n = 3). (**F**) Peptide dissociation from H2-D$^b$ (300 nM) after addition (arrow) of unlabeled competitor peptide (300 μM) without TAPBPR or in combination with the interface mutants TN6-TAPBPR and TN3-Ala-TAPBPR (1 μM each), respectively. A TAPBPR$^{wt}$-catalyzed peptide release reaction is shown as reference. The average dissociation rate constants in the presence of TN6 ($k_{off} = 2.53 \pm 0.30 \times 10^{-3}$ s$^{-1}$) and TN3-Ala ($k_{off} = 4.23 \pm 0.45 \times 10^{-3}$ s$^{-1}$) are shown in panel (**C**). Abbreviations: β2m: β2-microglobulin; MHC I hc: MHC I heavy chain; pMHC I: peptide-MHC I; mP: milli-polarization units; wt: wildtype; Tsn: tapasin; SL: scoop loop.

The online version of this article includes the following figure supplement(s) for figure 3:

**Figure supplement 1.** Catalyzed peptide displacement from H2-D$^b$ at low TAPBPR concentration.
**Figure supplement 2.** TAPBPR$^{wt}$-catalyzed displacement of high-affinity peptide from H2-D$^b$.

exhibited slightly reduced activity ($7.68 \pm 1.17 \times 10^{-3}$ $s^{-1}$) compared to the wt protein ($10.41 \pm 0.54 \times 10^{-3}$ $s^{-1}$), whereas TAPBPR$^{Tsn-SL}$ was slightly more active ($12.64 \pm 1.03 \times 10^{-3}$ $s^{-1}$) (*Figure 3B,C*). When we performed the experiment at a much lower TAPBPR concentration (75 nM), the TAPBPRs retained their activity, and the gradual activity differences between the variants remained (*Figure 3—figure supplement 1*). This suggests that TAPBPR$^{wt}$ and the scoop-loop mutants have similar affinities for H2-D$^b$. TAPBPR$^{wt}$ was even able to catalyze displacement of a high-affinity peptide from H2-D$^b$, although the catalytic effect was considerably smaller (1.8-fold acceleration) than for H2-D$^b$ loaded with the medium-affinity peptide (4.1-fold acceleration) (*Figure 3—figure supplement 2A,B*). In a second set of experiments, we analyzed peptide dissociation from the human MHC I allomorph HLA-A*02:01. Similar to the experiments with H2-D$^b$, in a peptide-rich environment (1000-fold molar excess of peptide), the highest catalytic activity towards HLA-A*02:01 was observed for TAPBPR$^{Tsn-SL}$, followed by TAPBPR$^{wt}$ and TAPBPR$^{\Delta SL}$; yet, the differences in activity between the three TAPBPRs were more pronounced, and the acceleration of the uncatalyzed peptide dissociation from HLA-A*02:01 ($1.90 \pm 0.04 \times 10^{-3}$ $s^{-1}$) by TAPBPR$^{Tsn-SL}$ ($26.31 \pm 2.59 \times 10^{-3}$ $s^{-1}$) and TAPBPR$^{wt}$ ($15.79 \pm 0.71 \times 10^{-3}$ $s^{-1}$) was significantly higher than for H2-D$^b$, while the activity of TAPBPR$^{\Delta SL}$ ($8.52 \pm 1.18 \times 10^{-3}$ $s^{-1}$) remained almost the same (*Figure 3D,E*).

The validity of our peptide exchange assay was confirmed by two interface mutants of TAPBPR$^{wt}$, TN3-Ala and TN6. The TN3 (E72K) and TN6 (E185K, R187E, Q189S, Q261S) mutants were initially described for Tsn to significantly reduce or abolish MHC I binding (*Dong et al., 2009*). The impact of the TN6 mutations on MHC I interaction was later confirmed for TAPBPR (*Morozov et al., 2016*). According to the TAPBPR-MHC I crystal structures (*Jiang et al., 2017*; *Thomas and Tampé, 2017a*), the residue in TAPBPR (E105) corresponding to the mutated residue in Tsn-TN3 forms a hydrogen bond with the swung-out Y84 of the MHC heavy chain, which is involved in coordinating the C-terminus of the peptide in liganded MHC. We reasoned that a mutation to Ala instead of Lys might increase the mutational effect and therefore generated the TN3-Ala mutant. Two of the mutated residues in TN6 (R210 and Q212) are part of the jack hairpin of TAPBPR and form several interactions with MHC I heavy-chain residues, while Q275 lies in the interface with the α2–1 helix and the β8 sheet in the floor of the MHC I binding groove. Consequently, TN3-Ala and TN6 displayed drastically reduced activity towards H2-D$^b$ in our peptide-exchange experiment, with peptide dissociation rate constants close to the value of the uncatalyzed reaction (*Figure 3C,F*). In summary, the results of our exchange assays demonstrate that under peptide-rich condition, the tested TAPBPR variants differ gradually in their displacement activity in an allomorph-dependent manner. But even the TAPBPR$^{\Delta SL}$ mutant lacking the scoop loop is still able to substantially accelerate peptide dissociation from MHC I.

## The scoop loop acts as an internal peptide competitor

In the TAPBPR-MHC I crystal structure, the scoop loop binds in the F pocket region of the MHC binding groove and appears to act as a surrogate for the peptide C terminus (*Thomas and Tampé, 2017a*). This notion is corroborated by our SEC analyses, which show that the scoop loop stabilizes peptide-free MHC I. We therefore wondered if the scoop loop impedes rebinding of displaced peptide and functions 'in cis' as a tethered, internal peptide competitor in the F pocket with extremely high effective concentration. To test this hypothesis, we modified the peptide exchange assay for H2-D$^b$ and HLA-A*02:01 by adding in a first step only TAPBPR without competitor peptide, which allowed us to monitor the change in free and bound fluorescent peptide under the influence of peptide rebinding in the presence of TAPBPR (*Figure 4A*). This condition mimics the physiological environment TAPBPR is operating in, where optimal replacement peptides are scarce. Strikingly, after addition of the different TAPBPRs to H2-D$^b$ loaded with fluorescent peptide, the polarization changes, which correspond to the changes in the ratio of free to bound peptide, diverged dramatically (*Figure 4B*). Peptide dissociation was most pronounced for TAPBPR$^{wt}$ with the native scoop loop, reaching ~ 60% peptide release, whereas only ~ 12% of the peptide population was released from H2-D$^b$ by TAPBPR$^{Tsn-SL}$, and almost no decrease in polarization was caused by TAPBPR$^{\Delta SL}$. Similar to our original peptide exchange assay (*Figure 3*), differences between the two MHC I allomorphs were observed: In comparison to H2-D$^b$, TAPBPR$^{Tsn-SL}$-induced peptide dissociation from HLA-A*02:01 was significantly stronger, approaching the level of peptide release induced by TAPBPR$^{wt}$ (*Figure 4—figure supplement 1A*). Peptide release was also peptide-dependent, as H2-

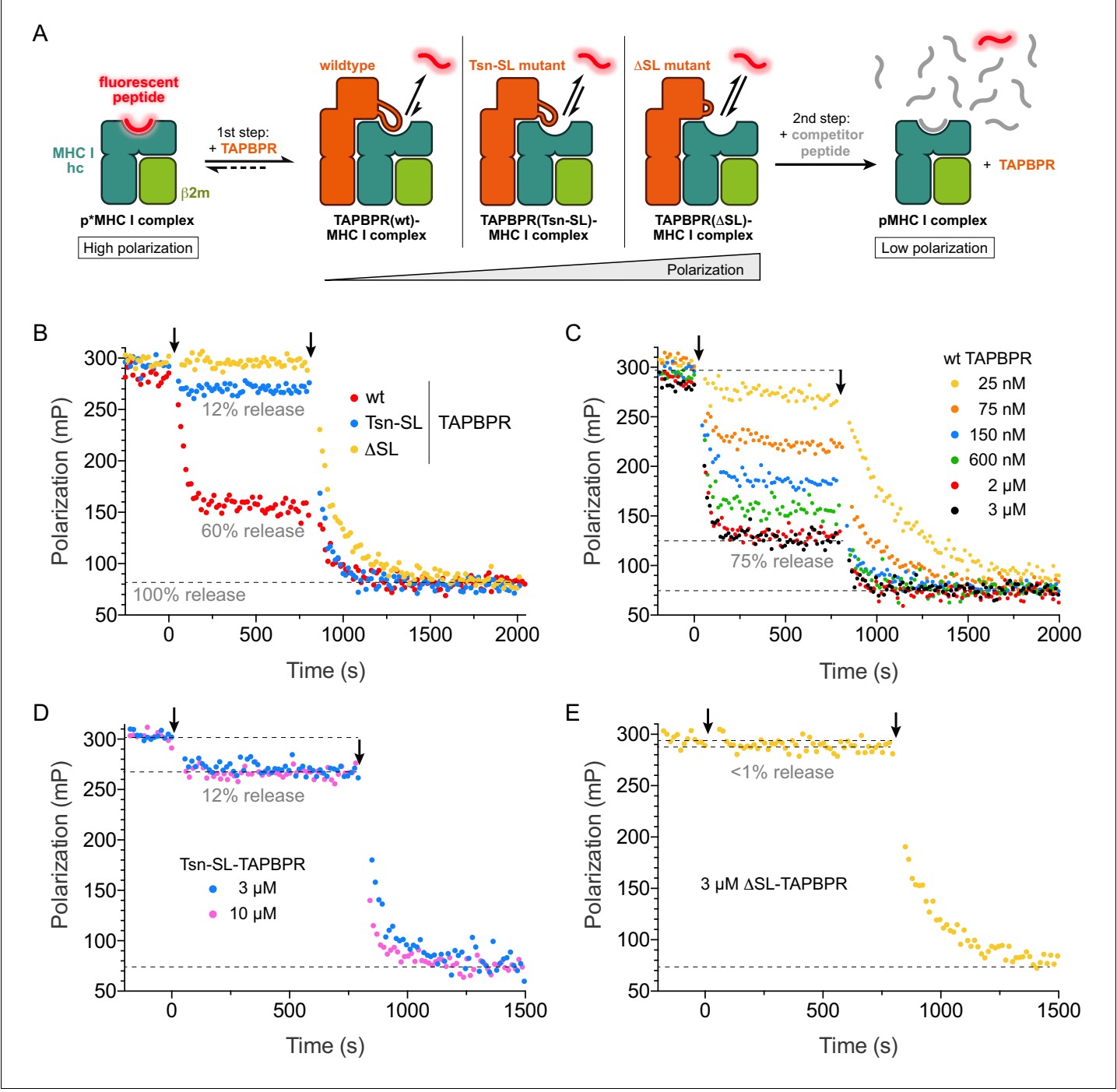

**Figure 4.** The scoop loop acts as a selectivity filter during peptide editing. (A) Schematic of two-step peptide exchange assay. (B) Peptide displacement from H2-D$^b$ (300 nM) loaded with fluorescently-labeled peptide (TQSC*NTQSI) was monitored by fluorescence polarization after addition of TAPBPR (1 μM, first arrow) and after subsequent addition of a 1000-fold molar excess of unlabeled high-affinity competitor peptide (ASNENMETM, 300 μM, second arrow). (C) Titration of peptide-loaded H2-D$^b$ (300 nM) with varying concentrations of TAPBPR$^{wt}$ (first arrow) and final addition of a 1000-fold molar excess of unlabeled high-affinity competitor peptide (300 μM, second arrow). (D) Peptide displacement from H2-D$^b$ (300 nM) loaded with fluorescently-labeled peptide monitored by fluorescence polarization after addition of 3 μM and 10 μM TAPBPR$^{Tsn-SL}$, respectively (first arrow), and after subsequent addition of a 1000-fold molar excess of unlabeled high-affinity competitor peptide (300 μM, second arrow). (E) Peptide displacement from H2-D$^b$ (300 nM) loaded with fluorescently-labeled peptide monitored by fluorescence polarization after addition of TAPBPR$^{ΔSL}$ (3 μM, first arrow) and after subsequent addition of a 1000-fold molar excess of unlabeled high-affinity competitor peptide (300 μM, second arrow). Data shown in (B)-(E) are representative of three independent measurements.

*Figure 4 continued on next page*

*Figure 4 continued*

The online version of this article includes the following figure supplement(s) for figure 4:

**Figure supplement 1.** The scoop loop acts as a selectivity filter during peptide editing.

D$^b$ loaded with a high-affinity peptide led to a significantly smaller decline in bound peptide (*Figure 3—figure supplement 2C*). After addition of competitor peptide (2$^{nd}$ step), the observed dissociation rate constants were in the same range as the values determined for the one-step experiment. Moreover, the level of released peptide after TAPBPR addition was titratable and reached saturation at 3 μM TAPBPR (*Figure 4C–E*, *Figure 4—figure supplement 1B*). Under the given conditions, TAPBPR$^{wt}$ was able to dissociate 70% (H2-D$^b$) and 80% (HLA-A*02:01) of total MHC I-associated peptide, respectively (*Figure 4C*, *Figure 4—figure supplement 1B*). These results suggest that the scoop loop interferes with re-binding of displaced peptide. It can only be completely dislodged from the MHC I binding pocket by a high-affinity peptide. The scoop loop thus acts as a crucial selectivity filter during peptide editing on MHC I.

## Discussion

Tsn and TAPBPR are MHC I-dedicated chaperones, which facilitate loading and selective exchange of antigenic peptides and thereby generate stable pMHC I complexes that shape a hierarchical immune response. The molecular underpinnings of their chaperone and peptide proofreading activities have only recently been uncovered by crystal structures of the TAPBPR-MHC I complex (*Jiang et al., 2017*; *Thomas and Tampé, 2017a*). Notably, one of the X-ray structures resolved a loop structure, termed the scoop loop, that is wedged into the F-pocket region of the empty MHC I binding groove and has been postulated to play an important role during peptide exchange (*Thomas and Tampé, 2017a*). Here, we show that the TAPBPR scoop loop is indeed critically important in chaperoning intrinsically unstable empty MHC I clients in a peptide-depleted environment. This is illustrated by the reduced chaperone activity of TAPBPR$^{Tsn-SL}$, which harbors the shorter Tsn scoop loop, and by the dramatically reduced lifetime of the TAPBPR$^{ΔSL}$ complex. In a peptide-rich, PLC-like environment, emulated by our one-step displacement experiments, the TAPBPR$^{Tsn-SL}$ mutant displays the highest activity, while TAPBPR$^{ΔSL}$ retains the ability to displace peptide. The latter observation appears to be in contrast to the study by Ilca et al. which found that TAPBPR with a mutated, but full-length scoop loop loses its ability to effectively mediate peptide dissociation (*Ilca et al., 2018*). In addition to stabilizing the chaperone-MHC I complex, we demonstrate that the TAPBPR scoop loop acts as an internal peptide competitor, and thus, as a selectivity filter in the discrimination between low- and high-affinity peptides. Although a direct competition appears to be the most obvious explanation for the effect on peptide rebinding, we cannot exclude that the scoop loop exerts its influence on peptide rebinding by an allosteric mechanism. The peptide-filtering activity seems to be allomorph-dependent for TAPBPR$^{Tsn-SL}$. Our current interpretation of this allomorph specificity is that the Tsn scoop loop interacts more strongly with the F-pocket region of HLA-A*02:01 and is therefore able to impede peptide rebinding more efficiently than in the case of H2-D$^b$. In contrast, TAPBPR$^{wt}$ shows a strong peptide release activity towards both MHC I allomorphs.

Based on the new insights, we propose the following model of TAPBPR-catalyzed peptide optimization on MHC I (*Figure 5*): The large concave surface formed by the N-terminal domain of TAPBPR mediates its initial encounter with a suboptimally-loaded MHC I, assisted by the C-terminal domain of TAPBPR, which contacts the α3 domain of the MHC I heavy chain and β2m. TAPBPR facilitates the release of low- to medium-affinity peptides primarily by widening the peptide-binding groove through the MHC I α2–1-helix, fastening the peptide-coordinating Tyr84, distorting the floor of the binding groove, and shifting the position of β2m (*Jiang et al., 2017*; *Thomas and Tampé, 2017a*). This remodeling is made possible by the intrinsic plasticity of MHC I molecules (*Bailey et al., 2015*; *Garstka et al., 2011*; *McShan et al., 2019*; *Natarajan et al., 2018*; *Thomas and Tampé, 2017b*; *van Hateren et al., 2017*; *van Hateren et al., 2015*; *Wieczorek et al., 2017*), and it appears to be induced primarily by structural elements of TAPBPR that lie outside the scoop loop. As a result, the TAPBPR$^{ΔSL}$ mutant lacking the scoop loop is still able to catalyze peptide displacement. Once the

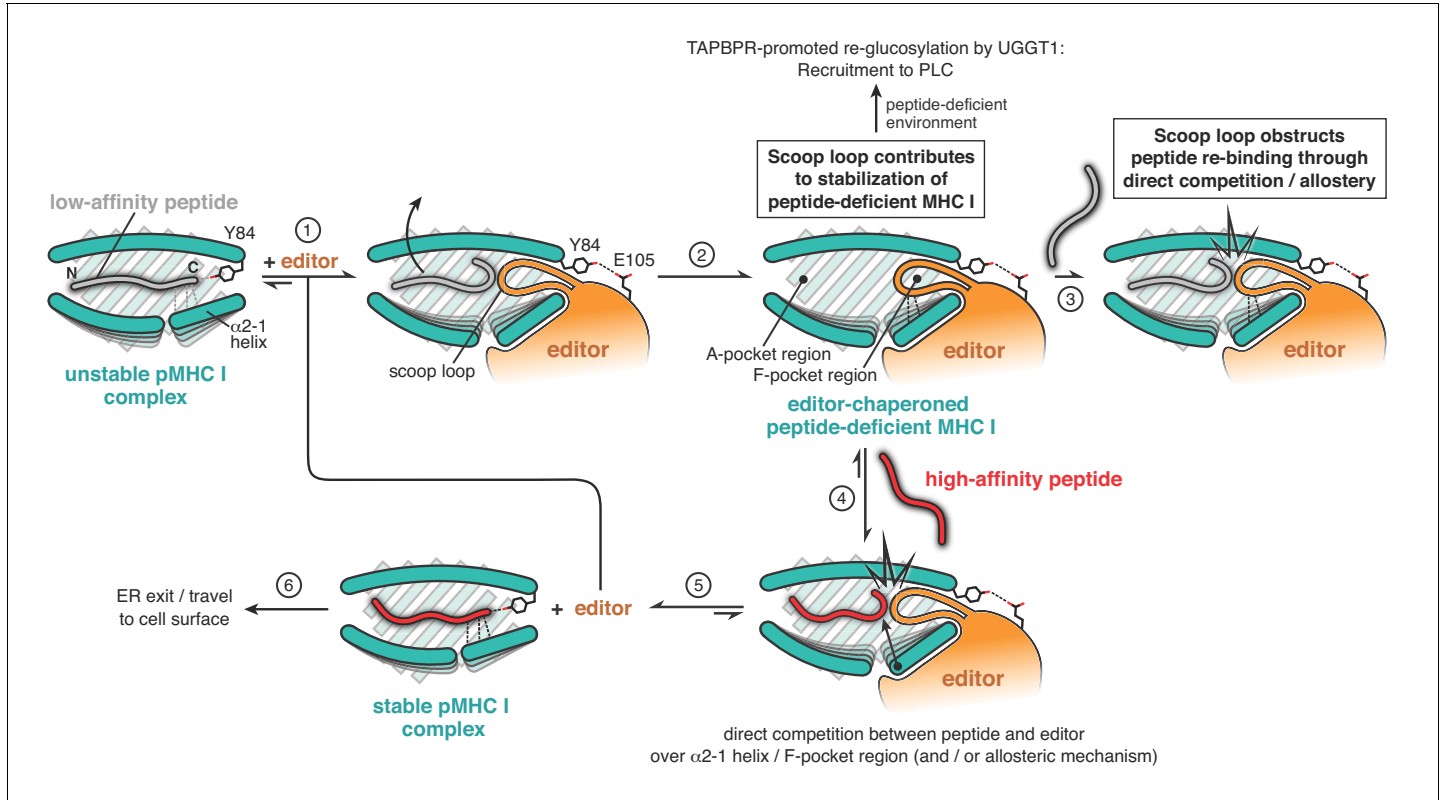

**Figure 5.** Proposed mechanistic functions of the scoop loop in catalyzed peptide proofreading. MHC I molecules bound to low-affinity peptide are recognized by the peptide editor (TAPBPR) (step 1). The editor lowers the peptide affinity of the suboptimally-loaded MHC I and induces dissociation of the low- to medium-affinity peptide (step 2). The scoop loop, which inserts into the F-pocket region of the peptide-binding groove, crucially contributes to the stabilization of the empty MHC I. In the absence of suitable peptides, empty MHC I clients are thereby held in a stable state until they can be loaded with an optimal epitope, for example in the PLC. Re-binding of the low-affinity peptide (step 3) is impeded by the scoop loop, through direct competition and/or via allosteric means. Only high-affinity peptides are able to compete with the editor over key regions of the peptide-binding groove (step 4) to eventually displace the scoop loop and the editor from the MHC I (step 5). The displaced editor is now ready for a new round of peptide selection, and the stable pMHC I complex is licensed to travel via the Golgi apparatus to the cell surface.

suboptimal peptide has been released, the scoop loop occupies the position of the peptide C terminus in the F-pocket region. The scoop loop thereby contributes to the stabilization of the peptide-deficient binding groove. Our two-step peptide exchange — mimicking a peptide-depleted environment — demonstrates that the scoop loop functions at the same time as a peptide selectivity filter by impeding re-binding of the replaced peptide, either through direct competition with the C terminus of the incoming replacement peptide or through an allosteric mechanism. Hence, the scoop loop contributes to the significant affinity decrease of incoming peptides for the MHC I groove in the presence of TAPBPR (*McShan et al., 2018*). Assuming a mode of direct competition, the replacement peptide would dock in the MHC I groove first with its N terminus, before it competes with the TAPBPR scoop loop over the F pocket region (*Hafstrand et al., 2019*; *Thomas and Tampé, 2017a*). Negative allosteric coupling between different parts of the MHC I molecule might play a role in the final release of TAPBPR (*McShan et al., 2018*). The shorter scoop loop in Tsn suggests that its selective pressure on the replacement peptide is weaker than in TAPBPR. Indeed, our fluorescence polarization and SEC analyses show that the tapasin scoop loop in TAPBPR$^{Tsn-SL}$ is less efficient in preventing re-binding of dissociated peptide.

Physiologically, these observations might be explained by the fact that Tsn functions within the PLC, a 'nanocompartment' characterized by an abundant and diverse supply of optimal peptides, reaching a bulk concentration of up to 16 μM before the TAP transporter is arrested by trans-inhibition (*Grossmann et al., 2014*). Moreover, Tsn is supported by other PLC chaperones in stabilizing empty MHC I clients. In contrast, TAPBPR operates as a single MHC I-dedicated chaperone outside the PLC in environments where the concentration of high-affinity peptides is drastically lower and

MHC I clients have to be stabilized in a peptide-receptive state for extended periods of time. Long-term stabilization of suboptimally-loaded or empty MHC I by TAPBPR also allows the major ER/*cis*-Golgi glycoprotein folding sensor UGGT1 (UDP-glucose:glycoprotein glucosyltransferase 1) to re-glucosylate the MHC I molecule in order to feed it back into the calnexin/calreticulin cycle and/or allow recruitment of the MHC I to the PLC (*Neerincx et al., 2017*; *Thomas and Tampé, 2019*). In conclusion, the evidence provided by our study indicates that the scoop loop is evolutionarily fine-tuned to enable Tsn and TAPBPR to accomplish their dual function as chaperone and proofreader in the specific subcellular location they operate in. By serving both as a stabilizing element and as selectivity filter in TAPBPR, the scoop loop influences peptide editing and impacts the repertoire of MHC I-associated epitopes presented on the cell surface.

# Materials and methods

**Key resources table**

| Reagent type (species) or resource | Designation | Source or reference | Identifiers | Additional information |
|---|---|---|---|---|
| Gene (human) | TAPBPR$^{wt}$ | PMID:29025996 | | lumenal domain |
| Gene (human) | TAPBPR$^{\Delta SL}$ | This study (*Figure 1C*, Materials and methods section) | | lumenal domain |
| Gene (human) | TAPBPR$^{Tsn-SL}$ | This study (*Figure 1C*, Materials and methods section) | | lumenal domain |
| Gene (human) | TAPBPR$^{TN3-Ala}$ | PMID:19119025 | | lumenal domain |
| Gene (human) | TAPBPR$^{TN6}$ | PMID:19119025 | | lumenal domain |
| Gene (human) | HLA-A*02:01 | This study (Materials and methods section) | | ectodomain |
| Gene (human) | β2-microglobulin | PMID:29025996 | | |
| Gene (mouse) | H2-D$^{b}$ | PMID:29025996 | | ectodomain |
| Strain, strain background (*Escherichia coli*) | DH10Bac | Thermo Fisher Scientific | 10361012 | chemically competent |
| Strain, strain background (*Escherichia coli*) | BL21(DE3) | Sigma-Aldrich | CMC0014 | chemically competent |
| Recombinant DNA reagent | pET-22 | Novagen/ Merck Millipore | 69744 | vector for protein expression in *E. coli* |
| Recombinant DNA reagent | pET-28 | Novagen/ Merck Millipore | 69864 | vector for protein expression in *E. coli* |
| Recombinant DNA reagent | pFastBacl-gp67 | PMID:29025996 | | transfer vector for Bac-to-Bac system |
| Cell line (*Spodoptera frugiperda*) | Sf9 | Thermo Fisher Scientific | 11496015 | |
| Cell line (*Spodoptera frugiperda*) | Sf21 | Thermo Fisher Scientific | 11497013 | |
| Peptide, recombinant protein | RGPGRAFJ*TI (photo-P18-I10) | PMID:26869717 | | J* denotes photo-cleavable amino acid |
| Peptide, recombinant protein | ASNENMETM | IEDB: epitope ID 4602 | | competitor peptide for H2-D$^{b}$ |

*Continued on next page*

*Continued*

| Reagent type (species) or resource | Designation | Source or reference | Identifiers | Additional information |
|---|---|---|---|---|
| Peptide, recombinant protein | FLPSDEEPYV | This study (Materials and methods section) | | competitor peptide for HLA-A*02:01 |
| Peptide, recombinant protein | TQSC*NTQSI | This study (Materials and methods section) | | C* denotes TAMRA-labeled Cys |
| Peptide, recombinant protein | FLPSDC*FPSF | This study (Materials and methods section) | | C* denotes TAMRA-labeled Cys |
| Peptide, recombinant protein | ASNC*NMETM | This study (Materials and methods section) | | C* denotes TAMRA-labeled Cys |
| Chemical compound, drug | TAMRA-5 maleimide | Thermo Fisher Scientific | T6027 | |
| Chemical compound, drug | TAMRA-6 C2 maleimide | Thermo Fisher Scientific | 48180 | |
| Chemical compound, drug | Fmoc-3-amino-3-(2-nitro) phenyl-propionic acid | Peptech | CAS #: 517905-93-0 | |
| Software, algorithm | Prism 6 | GraphPad Software | | |
| Software, algorithm | Fityk 1.3.1 | DOI: 10.1107/S0021889810030499 | | |
| Other | Superdex 200 Increase 10/300 | GE Healthcare | 28990944 | SEC column |
| Other | Superdex 200 Increase 3.2/300 | GE Healthcare | 28990946 | SEC column |
| Other | Superdex 75 Increase 3.2/300 | GE Healthcare | 29148723 | SEC column |
| Other | HiLoad Superdex 75 16/60 | GE Healthcare | 28989333 | SEC column |
| Other | Fluorolog-3 | Horiba Jobin Yvon | | spectro-fluorometer |
| Other | Äkta Purifier | GE Healthcare | | protein purification |
| Other | Agilent 1200 | Agilent | | analytical SEC |
| Other | Liberty Blue | CEM Corporation | | peptide synthesizer |
| Other | X-tremeGENE HP | Sigma-Aldrich | 6366236001 | transfection reagent |

## DNA constructs

The DNA constructs of human β2m, the ectodomain of mouse H2-D$^b$, and TAPBPR$^{wt}$ were identical to the ones previously described (*Thomas and Tampé, 2017a*), except for position 97 in TAPBPR$^{wt}$, which contained the native cysteine. The TAPBPR scoop loop mutants TAPBPR$^{Tsn-SL}$ and TAPBPR$^{\Delta SL}$ were generated by overlap extension PCR, the TN3-Ala and TN6 mutants were generated by site-directed mutagenesis. The TN3-Ala and TN6 mutants harbored the same mutations that were described for the corresponding mutants of Tsn (*Dong et al., 2009*), except that in TN3-Ala E105 was mutated to alanine. TAPBPR$^{Tsn-SL}$, TAPBPR$^{\Delta SL}$, TN3-Ala, and TN6 all contained the C97A mutation. Human HLA-A*02:01 (amino acids 1–278) was cloned into pET-28 (Novagen, Merck Millipore, Darmstadt, Germany) and ended in a C-terminal His$_6$-tag preceded by a linker (sequence: HE). The amino acid numbering of TAPBPR is based on the mature protein as defined by N-terminal sequencing (*Zhang and Henzel, 2004*).

## Protein expression

Human β2m and the ectodomains of mouse H2-D$^b$ and human HLA-A*02:01 were expressed as inclusion bodies in *Escherichia coli* BL21(DE3) as described before (*Rodenko et al., 2006*; *Thomas and Tampé, 2017a*). TAPBPR proteins were expressed in *Spodoptera frugiperda* (*Sf*21 or

*Sf9*) insect cells according to standard protocols for the Bac-to-Bac system (Thermo Fisher Scientific, Waltham, MA). A high-titer recombinant baculovirus stock was used to infect the insect cells at a density of $1.5–2.0 \times 10^6$ cells/mL, which were cultivated in Sf-900 III SFM medium (Thermo Fisher Scientific) at 28°C. The cell culture medium containing secreted TAPBPR was harvested 72 hr after infection.

## Refolding and purification of β2m

β2m was refolded by dialysis essentially as described previously (*Rodenko et al., 2006*) and purified by SEC on a Superdex 75 column (GE Healthcare, Piscataway, NJ) in HEPES-buffered saline (1xHBS: 10 mM HEPES pH 7.2, 150 mM NaCl). Purified protein was concentrated by ultrafiltration (Amicon Ultra 3 kDa MWCO, Merck Millipore).

## Peptide synthesis and labeling

The following peptides were used: the photo-cleavable peptide photo-P18-I10 (RGPGRAFJ*TI) (H2-D$^b$) [J*=3-amino-3-(2-nitro)phenyl-propionic acid], the non-fluorescent competitor peptides ASNEN-METM (H2-D$^b$) and FLPSDEEPYV (HLA-A*02:01), as well as the fluorescently labeled peptides TQSC*NTQSI (H2-D$^b$), FLPSDC*FPSF (HLA-A*02:01), ASNC*NMETM (H2-D$^b$) (C* denotes TAMRA-labeled cysteine). Non-natural peptide epitopes were designed based on their theoretical affinities according to the NetMHCpan server (*Jurtz et al., 2017*). While TQSC*NTQSI and FLPSDC*FPSF were constructed to have medium affinity (500–600 nM), ASNC*NMETM and the competitor peptides were designed to be high-affinity (8–10 nM) ligands. Peptides were synthesized using standard Fmoc solid-phase chemistry and purified by C$_{18}$ reversed-phase HPLC. The identity of peptides was verified either by matrix-assisted laser desorption/ionization mass spectrometry (MALDI-MS) or by electrospray ionization-mass spectrometry (ESI-MS). In order to site-specifically label peptides with fluorophores, 10.5 µM peptide were incubated with 26 µM TAMRA-5-maleimide (single isomer, Thermo Fisher Scientific) or TAMRA-6 C2 maleimide (Lumiprobe, Hannover, Germany) (used for labeling of FLPSDC*FPSF) overnight at 4°C. Labeled peptides were purified by C$_{18}$ reversed-phase HPLC, and their identity was confirmed by ESI-MS.

## Refolding and purification of MHC I allomorphs

H2-D$^b$ and HLA-A*02:01 were refolded from inclusion bodies by rapid dilution in the presence of purified β2m and peptide according to established protocols (*Rodenko et al., 2006*). Refolded MHC I complexes were purified by SEC (Superdex 200 Increase 10/300, GE Healthcare) in 1xHBS and concentrated by ultrafiltration (Amicon Ultra, Merck Millipore).

## Purification of TAPBPR proteins

TAPBPR proteins were purified from the insect cell culture medium by IMAC according to a protocol published earlier (*Thomas and Tampé, 2017a*), polished by SEC (Superdex 200 Increase 10/300, GE Healthcare) in 1xHBS, and concentrated by ultrafiltration (Amicon Ultra, Merck Millipore).

## Peptide exchange

Dissociation of fluorescently labeled peptide from MHC I was monitored at 23°C in 1xHBS by fluorescence polarization (Fluorolog-3 spectrofluorometer, Horiba Jobin Yvon, Bensheim, Germany) with $\lambda_{ex/em}$ of 530/560 nm. One-step and two-step dissociation assays were carried out with 300 nM MHC I loaded with TAMRA-labeled peptide, 1 µM TAPBPR, and 300 µM competitor peptide. Dissociation rate constants were determined in GraphPad Prism using a one-phase exponential decay regression.

## MHC I-chaperone complex formation

In the presence of purified TAPBPR (3 µM), photo-P18-I10-loaded H2-D$^b$ (10 µM) was irradiated with UV light (36 nm, 185 mW/cm$^2$, 120 s) on ice and afterwards incubated for 10 min at room temperature. Samples were subsequently centrifuged at 10,000xg for 10 min and analyzed by analytical SEC on a Superdex 75 (3.2/300) column (GE Healthcare). SEC runs were conducted in 1xHBS and monitored by absorbance at 280 nm. Chromatograms were deconvoluted into three Gaussian functions

using the program Fityk 1.3.1 (*Wojdyr, 2010*). The amount of complex was assessed by the area of the complex peak.

## TAPBPR-MHC I complex stability

Purified peptide-deficient TAPBPR$^{wt}$-H2-D$^b$, TAPBPR$^{Tsn-SL}$-H2-D$^b$, and TAPBPR$^{\Delta SL}$-H2-D$^b$ complexes were analyzed via analytical SEC either on a Superdex 75 (3.2/300) or a Superdex 200 (3.2/300) column (GE Healthcare) at a flow rate of 0.075 mL/min. A separate sample of purified TAPBPR$^{wt}$-H2-D$^b$ complex was incubated with a 100-fold molar excess of high-affinity peptide prior to re-analysis by SEC.

# Acknowledgements

We thank Christian Winter for help with peptide synthesis and analytical SEC. We thank Dr. Rupert Abele, Dr. Simon Trowitzsch, Andrea Pott, Inga Nold, and all members of the Institute for Biochemistry for discussion and comments. The support by the European Research Council (ERC Advanced Grant 789121 to RT) and the German Research Foundation (Reinhart Koselleck Project TA 157/12–1 to RT) is gratefully acknowledged.

# Additional information

### Funding

| Funder | Grant reference number | Author |
| --- | --- | --- |
| European Research Council | ERC_AdG 789121 | Robert Tampé |
| Deutsche Forschungsgemeinschaft | TA 157/12-1 | Robert Tampé |

The funders had no role in study design, data collection and interpretation, or the decision to submit the work for publication.

### Author contributions

Lina Sagert, Data curation, Formal analysis, Visualization, Methodology, Writing - original draft; Felix Hennig, Data curation, Formal analysis; Christoph Thomas, Conceptualization, Data curation, Formal analysis, Supervision, Validation, Visualization, Writing - original draft, Writing - review and editing; Robert Tampé, Conceptualization, Data curation, Formal analysis, Supervision, Funding acquisition, Validation, Investigation, Visualization, Writing - original draft, Project administration, Writing - review and editing

### Author ORCIDs

Christoph Thomas (iD) https://orcid.org/0000-0001-7441-1089
Robert Tampé (iD) https://orcid.org/0000-0002-0403-2160

### Decision letter and Author response

Decision letter https://doi.org/10.7554/eLife.55326.sa1
Author response https://doi.org/10.7554/eLife.55326.sa2

# Additional files

### Supplementary files

• Transparent reporting form

### Data availability

All data generated or analysed during this study are included in the manuscript and supporting files.

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
