## [Decision Letter]

**Acceptance summary:**

These careful experiments to analyze the role of the scoop loop in TAPBPR's chaperone activity during peptide loading onto the MHC I proteins will add to the general understanding of antigen presentation.

**Decision letter after peer review:**

Thank you for submitting your article "A loop structure allows TAPBPR to exert its dual function as MHC I chaperone and peptide editor" for consideration by *eLife*. Your article has been reviewed by three peer reviewers, and the evaluation has been overseen by a Reviewing Editor and Tadatsugu Taniguchi as the Senior Editor. The following individuals involved in review of your submission have agreed to reveal their identity: Malini Raghavan (Reviewer #1); Scheherazade Sadegh-Nasseri (Reviewer #2); Efstratios Stratikos (Reviewer #3).

The reviewers have discussed the reviews with one another and the Reviewing Editor has drafted this decision to help you prepare a revised submission.

Summary:

TAPBPR functions as a chaperone and peptide editor during the assembly of MHC class I molecules. Two different crystal structures of TAPBPR-MHC class I complexes have been solved (Jiang et al., 2017; Thomas and Tampe, 2017). While the two structures show overall similarities in the interaction modes, the location of the scoop loop (approximately residues 22-25 of TAPBPR) in the F-pocket region of peptide-free MHC class I have been in debate (Natarajan et al., Crit Rev Biochem Mol Biol., 2019). More recent studies support a model in which the scoop loop plays an important role in TAPBPR-mediated peptide dissociation from MHC class I (Ilca et al., 2018). Using purified human and mouse MHC class I molecules, and purified TAPBPR and its scoop loop mutant, the new study by Sagert et al. further analyzes the role of the scoop loop in TAPBPR function. Overall, this is a very well written manuscript that presents well-thought experiments and interesting results. The experiments are well-performed and controlled, and the results are generally clear-cut, under the tested conditions. Concerns with the manuscript relate to some of the experimental interpretations and conclusions leading to the model.

Essential revisions:

1) Figure 2 nicely presents the formation of TAPBPR/H2-D^b^ and how mutating the scoop loop can affect the relative amount of complex formed. While the experiment is highly informative, I am not persuaded that it accurately reports kinetic instability as the authors claim. Rather it is equally possible that it reports thermodynamic destabilisation. Given the reported interactions of the scoop loop with the MHC I peptide-binding groove, lack of the scoop loop or sequence substitution would be expected to weaken the interaction, something that could explain the observations in Figure 2. Can the authors more directly measure the affinity of TAPBPR for MHC I? Even if this is not easily achievable, I think the authors should reconsider their claims for kinetic instability, since deconvolution of kinetics vs. thermodynamics is probably impossible by this particular experiment.

2) Given the result presented in Figure 2, how do the authors explain the finding that scoop loop variants largely retain their ability to catalyse peptide dissociation (Figure 3)? Is it because the scoop loop interactions with MHC are secondary to the overall TAPBPR-MHC interactions? If the scoop loop variants have a reduced affinity for MHC, would an effect be more obvious if experiments in Figure 3 are repeated at lower concentrations?

3) Relative to a TAPBPR chimeric construct containing the shorter scoop loop of tapasin, the wild type TAPBPR is shown to form more stable (higher affinity complexes) with MHC I (Figure 2), although the peptide dissociation function of the wild type TAPBPR is less efficient than the chimera with the tapasin scoop loop (Figure 3). The stronger peptide dissociation effect of a TAPBPR with the tapasin scoop loop compared with TAPBPR^wt^ should be interpreted, explicitly discussed, and taken into account in the overall model of Figure 5.

4) Notably, mutations of other key TABBPR residues at the TAPBPR-MHC class I interface (TN3 and TN6) are shown in the representative figure to have more significant effects on peptide dissociation than a TAPBPR mutant in which the scoop loop was mutated to a 3xGly sequence, suggesting a dominant role for TN3 and TN6-mediated interactions in inducing peptide dissociation. The dominant role of TN3 and TN6-mediated interactions in inducing peptide dissociation should be acknowledged (including within the model), and their quantifications shown in the same way as Figure 3C and 3E.

5) In a variant of the peptide dissociation experiment, peptide dissociation was measured in the presence of the three TAPBPR variants, but in the absence of excess unlabeled peptide (Figure 4 and Figure 4—figure supplement 1). Under this set of conditions, wild type TAPBPR was more efficient at inducing peptide dissociation from H2-D^b^ than the two scoop loop mutants, whereas wild type TAPBPR and the tapasin scoop loop mutant appear to have similar efficiencies towards peptide dissociation from HLA-A2. These data are used to argue for a model in which the scoop loop directly interferes with rebinding of a displaced peptide (internal competitor). However, concerns with these interpretations are:

a) Other models (such as different allosteric effects induced by the three scoop loop variants of TAPBPR) could explain the varying degrees of inhibition of rebinding of dissociated peptides by the TAPBPR variants, which are also predicted to have different affinities for MHC class I. Model validation will need data that can more clearly sort out the difference between an internal competitor model and alternative allosteric effects-based models. If the differences cannot be easily sorted out, at minimum, the model of Figure 5 should be changed to indicate that both the internal competitor model and allosteric model are likely to be relevant.

b) The effects of the three TAPBPR constructs on inhibition of peptide rebinding vary between HLA-A2 and H2-D^b^ (comparing Figure 4 with Figure 4—figure supplement 1). The variable effects of tapasin scoop loop on D^b^ vs. HLA-A2 should be expanded, interpreted, and explicitly discussed.

6) Does the experiment in Figure 4 report a reduced "functional" ability of the scoop loop mutants to remove the peptide from MHC or a reduced affinity of the TAPBPR variants for MHC? This difference is relevant to our understanding of the distinct functionality of the scoop loop (versus being a hinge that makes additional interactions with the MHC). Perhaps the authors could test this by titrating the less active TAPBPR^Tsn-SL^ to higher concentrations to see if they can get similar results to the TAPBPR^wt-SL^.

7) The importance of the scoop loop has recently been demonstrated in the study by Ilca et al. (2018), but the latter paper is only referenced to highlight the differences in conclusions, rather than many other overall similarities in findings. The Ilca et al. paper should be referenced within the Introduction for better acknowledgement of prior contributions to the same questions, and similarities with the present study pointed out in addition to the differences-within the Discussion.

8) Finally, for readers and for the field, this section in the Introduction is very confusing "Furthermore, one of the two TAPBPR-MHC I complex structures revealed a remarkable structural feature in TAPBPR named the scoop loop (Thomas and Tampe, 2017). […] The scoop loop occupies a position that is incompatible with high-affinity peptide binding and displaces or coordinates several key MHC I residues, including Y84, T143, K146, and W147, which are responsible for binding the C terminus of the peptide." The authors should rephrase to clarify what has been discovered regarding the scoop loop previously, what is in debate, and specify the open questions that this study will address.

9) As noted above, model validation will need data that can more clearly sort out the difference between an internal competitor model and alternative allosteric effects-based models. Additional higher resolution structural data, if possible to acquire within two months, may be helpful because the loop was modeled as inserting into the F-pocket in one of the structures (Thomas and Tampe, 2017) but not in the second (Jiang et al., 2017).

Overall, this is a well-executed study, but the data do not integrate well with the proposed model. In line with the above points, the data are more consistent with a relatively minor (and possibly MHC-context dependent) contribution of the scoop loop to peptide editing, rather than the current emphasis of a critical role. This message should be evident within the Abstract and throughout the manuscript.

---

## [Author Response]

Essential revisions:1) Figure 2 nicely presents the formation of TAPBPR/H2-D^b^ and how mutating the scoop loop can affect the relative amount of complex formed. While the experiment is highly informative, I am not persuaded that it accurately reports kinetic instability as the authors claim. Rather it is equally possible that it reports thermodynamic destabilisation. Given the reported interactions of the scoop loop with the MHC I peptide-binding groove, lack of the scoop loop or sequence substitution would be expected to weaken the interaction, something that could explain the observations in Figure 2. Can the authors more directly measure the affinity of TAPBPR for MHC I? Even if this is not easily achievable, I think the authors should reconsider their claims for kinetic instability, since deconvolution of kinetics vs. thermodynamics is probably impossible by this particular experiment.

This is indeed an important point. Directly determining the affinity of TAPBPR for peptide-deficient MHC I, as a measure of chaperone activity, is very difficult, as peptide-deficient MHCs are intrinsically unstable and will precipitate. We tried to determine K_d_ values via microscale thermophoresis using MHC I loaded with a photo-cleavable peptide; however, neither T-jump nor thermophoresis delivered a useful signal upon complex formation. In addition, we also aimed to determine *k_off_*for the complexes using fluorescently-labeled TAPBPRs, but the scoop loop mutants turned out to be unstable upon fluorophore attachment. This is why we resorted to SEC analysis using unlabeled proteins. We cannot exclude different equilibrium dissociation constants for the complexes. However, our re-injection experiments (Figure 2—figure supplement 1; SEC traces for the two TAPBPR scoop-loop mutants have now been added) clearly show that the three complexes have different kinetic stabilities. Furthermore, we have repeated the peptide displacement experiments at a much lower TAPBPR concentration (75 nM) (new Figure 3—figure supplement 1) and observed gradual activity differences between the three TAPBPR constructs that were similar to the previous measurements at 1 µM, suggesting that the TAPBPRs have affinities for (suboptimally-loaded, and most likely empty) MHC I that are at least in the same range.

2) Given the result presented in Figure 2, how do the authors explain the finding that scoop loop variants largely retain their ability to catalyse peptide dissociation (Figure 3)? Is it because the scoop loop interactions with MHC are secondary to the overall TAPBPR-MHC interactions? If the scoop loop variants have a reduced affinity for MHC, would an effect be more obvious if experiments in Figure 3 are repeated at lower concentrations?

Initially, we chose the high TAPBPR concentrations on purpose to test if the mutants are still active in catalyzing peptide displacement. We have now repeated the experiment at a much lower TAPBPR concentration (75 nM) (new Figure 3—figure supplement 1) and observed the same relative catalytic activities between the three TAPBPR constructs as before (at 1 µM), suggesting that the TAPBPRs have similar affinities for (suboptimally-loaded, and most likely empty) MHC I. Then, why are the different complex stabilities not reflected in the one-step peptide-displacement assay? We think there might be two reasons: First, the chaperone activity, i.e. the ability to stabilize peptide-free MHC I for a prolonged period of time, does not come into effect in our displacement assay, as the assay is carried out in the presence of a 1000-fold molar excess of high-affinity competitor peptide. Secondly, the scoop loop does not appear to be of primary importance for the displacement step in peptide exchange. Other TAPBPR-mediated, scoop loop-independent changes might be more important for induced peptide dissociation (widening of MHC binding groove, salt bridge to Y84, distortion of groove floor, shift of β2m).

3) Relative to a TAPBPR chimeric construct containing the shorter scoop loop of tapasin, the wild type TAPBPR is shown to form more stable (higher affinity complexes) with MHC I (Figure 2), although the peptide dissociation function of the wild type TAPBPR is less efficient than the chimera with the tapasin scoop loop (Figure 3). The stronger peptide dissociation effect of a TAPBPR with the tapasin scoop loop compared with TAPBPR^wt^ should be interpreted, explicitly discussed, and taken into account in the overall model of Figure 5.

As explained in our answer to point 2, we do not think that the chaperone activity (Figure 2) can be directly correlated with the activity in our peptide-displacement assay (Figure 3). Currently, we do not have an explanation for the slightly higher activity of the TAPBPR^Tsn-SL^ variant. However, our aim of the displacement measurements was to demonstrate that the two scoop-loop variants are still generally able to catalyze peptide dissociation. Notably, the conditions of the assay (high concentration of competitor peptide) resemble the physiological environment tapasin is operating in, i.e. the peptide-rich environment of the PLC. The tapasin scoop loop might thus be expected to operate optimally under these experimental conditions.

4) Notably, mutations of other key TABBPR residues at the TAPBPR-MHC class I interface (TN3 and TN6) are shown in the representative figure to have more significant effects on peptide dissociation than a TAPBPR mutant in which the scoop loop was mutated to a 3xGly sequence, suggesting a dominant role for TN3 and TN6-mediated interactions in inducing peptide dissociation. The dominant role of TN3 and TN6-mediated interactions in inducing peptide dissociation should be acknowledged (including within the model), and their quantifications shown in the same way as Figure 3C and 3E.

It is beyond debate that the Y84-coordinating glutamate in TAPBPR (mutated in the TN3 mutant) and residues mutated in TN6 are crucial for TAPBPR-catalyzed peptide displacement. Catalysis of peptide displacement by TAPBPR involves several induced changes in the MHC I, including a stabilized open conformation of the peptide binding groove, distortion of the groove floor, and displacement of β2m. These changes are mediated by parts of the TAPBPR molecule that are outside the scoop loop and comprise the greater part of the TAPBPR-MHC I interface. Especially, the TN6 mutations affect the core of the TAPBPR-MHC I interface and are therefore expected to drastically reduce displacement activity. We definitely do not claim that the scoop loop is more important for peptide displacement than the aforementioned aspects of catalysis. In the current manuscript we focus solely on the role of the scoop loop, and the main purpose of the experiments shown in Figure 3 was to demonstrate that our two scoop-loop variants are still able to catalyze peptide dissociation.

We have now included bar diagrams for the TN3 and TN6 measurements in Figure 3C.

5) In a variant of the peptide dissociation experiment, peptide dissociation was measured in the presence of the three TAPBPR variants, but in the absence of excess unlabeled peptide (Figure 4 and Figure 4—figure supplement 1). Under this set of conditions, wild type TAPBPR was more efficient at inducing peptide dissociation from H2-D^b^ than the two scoop loop mutants, whereas wild type TAPBPR and the tapasin scoop loop mutant appear to have similar efficiencies towards peptide dissociation from HLA-A2. These data are used to argue for a model in which the scoop loop directly interferes with rebinding of a displaced peptide (internal competitor). However, concerns with these interpretations are:a) Other models (such as different allosteric effects induced by the three scoop loop variants of TAPBPR) could explain the varying degrees of inhibition of rebinding of dissociated peptides by the TAPBPR variants, which are also predicted to have different affinities for MHC class I. Model validation will need data that can more clearly sort out the difference between an internal competitor model and alternative allosteric effects-based models. If the differences cannot be easily sorted out, at minimum, the model of Figure 5 should be changed to indicate that both the internal competitor model and allosteric model are likely to be relevant.

We have modified the manuscript and the text of Figure 5 to include the possibility of allosteric effects.

b) The effects of the three TAPBPR constructs on inhibition of peptide rebinding vary between HLA-A2 and H2-D^b^ (comparing Figure 4 with Figure 4—figure supplement 1). The variable effects of tapasin scoop loop on Db vs. HLA-A2 should be expanded, interpreted, and explicitly discussed.

We now explicitly describe the observed allomorph-specific activity and included a possible interpretation of this phenomenon.

6) Does the experiment in Figure 4 report a reduced "functional" ability of the scoop loop mutants to remove the peptide from MHC or a reduced affinity of the TAPBPR variants for MHC? This difference is relevant to our understanding of the distinct functionality of the scoop loop (versus being a hinge that makes additional interactions with the MHC). Perhaps the authors could test this by titrating the less active TAPBRP^Tsn-SL^ to higher concentrations to see if they can get similar results to the TAPBPR^wt-SL^.

We clearly show that the differences between wildtype TAPBPR and the scoop loop mutants persist after saturation (see titrations with increasing TAPBPR concentrations in Figure 4C and D, and the saturating concentration in Figure 4E). This demonstrates that the differences are not due to reduced affinities of the mutants.

7) The importance of the scoop loop has recently been demonstrated in the study by Ilca et al. (2018), but the latter paper is only referenced to highlight the differences in conclusions, rather than many other overall similarities in findings. The Ilca et al. paper should be referenced within the Introduction for better acknowledgement of prior contributions to the same questions, and similarities with the present study pointed out in addition to the differences-within the Discussion.

We now reference the Ilca et al. paper in the Introduction and contrast its cell-based experimental approach to our in vitro approach using defined, purified protein components.

8) Finally, for readers and for the field, this section in the Introduction is very confusing "Furthermore, one of the two TAPBPR-MHC I complex structures revealed a remarkable structural feature in TAPBPR named the scoop loop (Thomas and Tampe, 2017). […] The scoop loop occupies a position that is incompatible with high-affinity peptide binding and displaces or coordinates several key MHC I residues, including Y84, T143, K146, and W147, which are responsible for binding the C terminus of the peptide." The authors should rephrase to clarify what has been discovered regarding the scoop loop previously, what is in debate, and specify the open questions that this study will address.

We have simplified and rephrased the paragraph to describe the current state of knowledge and open questions.

9) As noted above, model validation will need data that can more clearly sort out the difference between an internal competitor model and alternative allosteric effects-based models. Additional higher resolution structural data, if possible to acquire within two months, may be helpful because the loop was modeled as inserting into the F-pocket in one of the structures (Thomas and Tampe, 2017) but not in the second (Jiang et al., 2017).

In contrast to our TAPBPR-H2-D^b^ structure (new Figure 1B showing electron density of scoop loop, contoured at 0.8 σ), the TAPBPR-H2-D^d^ complex (Jiang et al., 2017) does indeed lack density for the scoop loop. This is most likely due to the presence of a disulfide-linked 5-mer peptide in the binding groove of the H2-D^d^ construct used by Jiang et al., which interferes with stable scoop-loop binding. (Electron density for this covalently bound 5-mer peptide is also lacking, probably due to mobility). Higher-resolution crystallographic data might provide more detailed insights into the binding mode of the scoop loop, but the given time frame for revision (2 months) is definitely too short to obtain these data. It has taken us years of very focused efforts and numerous crystals to obtain the TAPBPR-H2-D^b^ structure at the current resolution.

Overall, this is a well-executed study, but the data do not integrate well with the proposed model. In line with the above points, the data are more consistent with a relatively minor (and possibly MHC-context dependent) contribution of the scoop loop to peptide editing, rather than the current emphasis of a critical role. This message should be evident within the Abstract and throughout the manuscript.

We have slightly toned down our statements about the role of the scoop loop in peptide editing. However, we still think that the observed effects on chaperone activity and peptide rebinding are striking.